# Formation of Food Grade Microemulsion with Rice Glycosphingolipids to Enhance the Oral Absorption of Coenzyme Q10

**DOI:** 10.3390/foods8100502

**Published:** 2019-10-15

**Authors:** Hiromasa Uchiyama, Jisoon Chae, Kazunori Kadota, Yuichi Tozuka

**Affiliations:** Osaka University of Pharmaceutical Sciences, 4-20-1 Nasahara, Takatsuki, Osaka 569-1094, Japan; uchiyama@gly.oups.ac.jp (H.U.); e13338@gap.oups.ac.jp (J.C.); kadota@gly.oups.ac.jp (K.K.)

**Keywords:** rice glycosphingolipids, microemulsion, freeze dry, coenzyme Q10, oral absorption

## Abstract

The purpose of this study is to examine the possible use of rice glycosphingolipids (RGSLs) as an emulsifier to form food grade microemulsions (mean particle size, 10–20 nm) and improve the absorption of CoQ10 with a poor solubility property by prepared emulsion. Because RGSLs could act as an auxiliary emulsifying agent, its addition to the surfactant/oil mixture decreased the emulsion’s particle size. This suggests that RGSLs exist between the water and oil phases to decrease oil droplet size via reduced interfacial tension. CoQ10-loaded microemulsion was also successfully prepared with RGSLs and powdered after freeze-drying with a cryoprotectant. CoQ10’s solubility in freeze-dried particles was dramatically improved compared to that of CoQ10 powder. Moreover, oral absorption of CoQ10 was significantly enhanced when administered via CoQ10-loaded microemulsion. The area under the plasma concentration–time curve for the microemulsion improved up to seven-fold compared to CoQ10 powder. The use of RGSLs could, therefore, be an effective processing technique for improving CoQ10’s solubility and absorption.

## 1. Introduction

Coenzyme Q10 (CoQ10), existing in the inside of the inner mitochondrial membrane, functions as the production source of cellular energy and adenosine triphosphate. [1]. Many researchers have reported the beneficial effects of CoQ10 which include its antioxidant, anti-inflammatory, and anti-atherogenic activities [2,3,4]. In addition, CoQ10 intake has been reported to improve the symptoms of early Parkinson’s disease and Huntington’s disease [5]. One of the most prominent setbacks for the application of CoQ10 as a supplement is its poor dispersibility and solubility in polar solvents such as water and ethanol. Terao et al. reported that the aqueous solubility of CoQ10 is less than 0.1 μg/mL [6]. Hence, the absorption of CoQ10 from the gastrointestinal tract is limited by its poor solubility and large molecular weight (863.34 g/moL) [7]. Oil-based formulations are available on the market as nutritional supplements to improve the absorption of CoQ10. In oil-based formulations, oil-in-water (O/W) emulsions encapsulated various poorly-soluble compounds, with a dispersed oil phase in aqueous solutions, are generally prepared by the apparatus, such as a high-pressure homogenizer and extruder, to supply energy that increases the water/oil interfacial area [8,9,10]. The particle size of emulsions prepared by these apparatuses shows the range from submicron to nano with a narrow particle distribution. [11,12]. Emulsions exhibiting a particle size less than 200 nm can be prepared by spontaneous emulsification without the use of an emulsifying apparatus [13,14]. Gulotta et al. reported that a nanoemulsion less than 100 nm, including ω-3 fatty acids, can be prepared using a spontaneous emulsification method [15]. On the other hand, it is important to select appropriate food additives to achieve spontaneous emulsification. Lipid mixtures with higher hydrophilic-lipophilic balance (HLB) surfactants, oils, and/or co-surfactants lead to the formation of nano-sized emulsion depending on their component ratios. Polysorbate 80 and d-α-tocopherolpolyethylene glycol 1000 succinate, a non-ionic emulsifier, have been examined as high HLB surfactants to prepare nano-sized emulsions [13,16]. The emulsion prepared with these surfactants exhibited a particle size lower than 100 nm because of their high emulsifying potential. Lecithin, a mixture of glycerophospholipid, can also be used as an emulsifier [17,18]. Zhou et al. reported that an increase in the concentration of soybean lecithin, when used as an emulsifier, resulted in a decrease in particle size [19]. Natural biocompatible compounds such as lecithin are expected to function as an emulsifier. 

Glycosphingolipids (GSLs) are essential components existing in mammalian cell membrane, particularly at the cell surface [20]. GSLs are amphipathic molecules that consist of a ceramide lipid moiety with an *N*-acylated sphingosine group and glucose or galactose linked to the primary hydroxy group of the sphingosine moiety [21]. GSLs have been studied in various areas because of their superior effects, such as improvement of the immune system, barrier recovery, and skin moisturizing property [22,23]. However, it is difficult to process GSLs as a nutritional supplement drink and an oil-based formulation because of their low solubility in water or oil alone. On the other hand, GSLs may have potential as an emulsifier like lecithin because they possess an amphiphilic moiety and are constituents of the cell membrane. In this study, rice glycosphingolipids (RGSLs) were investigated as an additive to prepare a nano-sized O/W emulsion containing CoQ10. The chemical structure of RGSLs is shown in Figure 1. The main composition of the fatty acid and sphingoid base in RGSLs was C20:0 and d18:2, respectively. Only a few studies have reported the application of RGSLs as an emulsifier.

The purpose of this study was to examine the possible use of RGSLs as an emulsifier and the effect of RGSLs addition to emulsion formation and enhance oral absorption of CoQ10 using prepared emulsion. Polysorbate 80 and Caprylic/capric triglyceride (TG) were used as a surfactant and oil, respectively. CoQ10 was used as a functional compound that displays poor water solubility. Because RGSLs do not dissolve in water and oil alone, CoQ10-loaded emulsion with RGSLs was prepared via hydration after evaporation. The powder containing CoQ10-loaded emulsion was prepared by the freeze-drying method with cryoprotectants. Furthermore, its re-dispersibility and solubility were evaluated using distilled water and biorelevant media. The pharmacokinetics of CoQ10 in a CoQ10-loaded emulsion were compared to those of untreated CoQ10 powder after oral administration to rats.

## 2. Materials and Methods 

### 2.1. Materials

RGSLs was supplied by Okayasu Shoten Co., Ltd (Saitama, Japan). Polysorbate 80 and TG were supplied by Nikko Chemicals Co., Ltd (Tokyo, Japan). Coenzyme Q10 (CoQ10) was a gift from Nisshin Pharma Inc. (Tokyo, Japan) and Kyowa Hakko Bio Co., Ltd. (Tokyo, Japan). D-mannitol and sucrose were purchased from Wako Pure Chemical Industries (Osaka, Japan). All other chemicals and solvents were either of reagent grade or high-performance liquid chromatography (HPLC) grade and were used without further purification. 

### 2.2. Particle Size Measurement

Particle sizes less than 1 μm were measured with Nanotrac UPA by dynamic light scattering (UPA-UT151; MicrotracBEL, Osaka, Japan), while sizes more than 1 μm were measured with Microtrac HRA by laser diffraction analysis (HRA-9320, MicrotracBEL). This measurement was performed in triplicate, and three times of data were averaged. A value of particle size was represented as volume-based particle size. 

### 2.3. Preparation of RGSLs, Polysorbate 80, or RGSLs/Polysorbate 80 particles

RGSLs (0.01, 0.1 and 1 mg), polysorbate 80 (210 mg), or RGSLs (50 mg) and Polysorbate 80 (210 mg) were dissolved in 10 mL of ethanol/acetone (50/50, v/v), and then processed by a rotary evaporator (R-3; Buchi, Tokyo, Japan) with a pressure of 102 mbar in a 50 °C water bath. The mixture of ethanol/acetone was removed via adequate evaporation. Distilled water (8 mL) was added to the evaporated samples and sonicated for 10 min. The particle size was determined by Nanotrac UPA or Microtrac HRA as described in Section 2.2.

### 2.4. Preparation of Emulsion Without CoQ10

RGSLs (0–90 mg), polysorbate 80 (0–280 mg), and TG (70–140 mg) were dissolved in 10 mL of ethanol/acetone (50/50, v/v) and processed by a rotary evaporator (R-3; Buchi) with pressure of 102 mbar in a 50 °C water bath. The mixture of ethanol/acetone was removed by adequate evaporation. Distilled water (2–16 mL) was added to the evaporated samples with a vortex and sonicated for 10 min. Particle size was determined by a Nanotrac UPA or Microtrac HRA as described in Section 2.2.

### 2.5. Preparation of CoQ10-Loaded Emulsion

Polysorbate 80 (210 mg), RGSLs (50 mg), TG (100 mg), and CoQ10 (10–40 mg) were dissolved in 10 mL of ethanol/acetone (50/50, v/v) and processed by a rotary evaporator (R-3; Buchi) with a 102 mbar pressure in a water bath at 50 °C. The mixture of ethanol/acetone was removed via adequate evaporation. Distilled water (8 mL) was added to evaporated samples with vortex and then sonicated for 10 min. Particle size was determined by a Nanotrac UPA as described in Section 2.2.

### 2.6. Encapsulation Efficiency of CoQ10

CoQ10-loaded emulsions were centrifuged at 6000*g* for 10 min (1524, BM Equipment Co., Ltd., Tokyo Japan). The concentration of CoQ10 in the supernatant was determined using an HPLC (SPD-10A, Shimadzu Co., Ltd., Kyoto, Japan) with an ENDURO C18 (5 μm, 150 mm × 4.6 mm, SGE Analytical Science) column. The mobile phase consisted of ethanol and methanol (70:30, v/v). The flow rate was controlled at 1.0 mL/min with a 30-μL injection volume. CoQ10 was eluted at 40 °C and quantified at 275 nm. The encapsulation efficiency of CoQ10 was determined by the following equation: (1)$$Encapsulation\;efficiency\;(\%)=\;\frac{\mathrm{CoQ}10\;\mathrm{concentration}\;\mathrm{in}\;\mathrm{supernatant}\;}{\mathrm{Total}\;\mathrm{CoQ}10\;\mathrm{concentration}}\times 100$$

The CoQ10-loaded emulsion was kept in a water bath at 37 °C for 4 weeks. Particle size and encapsulation efficiency of CoQ10 were evaluated at 1, 2, and 4 weeks. The particle size of the emulsion was measured using Nanotrac UPA. The encapsulation efficiency of CoQ10 was determined by HPLC as described in Section 2.6. 

### 2.7. Powder Preparation

The powder containing CoQ10-loaded emulsion was prepared with mannitol or sucrose by the freeze-drying method. A double volume of mannitol or sucrose for components of CoQ10-loaded emulsion, except water, was added and then frozen at −40 °C for 30 min in a circulation bath (PFR-1000, Tokyo Rikakikai Co., Ltd., Tokyo, Japan). Lyophilization was performed at around 5 Pa for 24 h using a freeze-drier (FDU-830 freeze dryer, Tokyo Rikakikai Co., Ltd.). 

### 2.8. Morphology of Freeze-Dried Particles

The morphology of samples was observed by scanning electron microscopy (SEM) (TM3030, HITACHI, Tokyo, Japan). Morphology observations were performed at an acceleration voltage of 15 kV. Samples were attached to carbon sticky tape mounted on SEM stubs. Prior to observation, samples were sputtered with a thin layer of platinum under vacuum.

### 2.9. Re-Dispersibility of Freeze-Dried Particles 

The freeze-dried powder was re-dispersed in 5 mL of distilled water to achieve the same concentration as that before freeze-drying. The particle size of re-dispersed CoQ10-loaded microemulsion was measured by Nanotrac UPA or Microtrac HRA as described in 2.2.

### 2.10. Solubility Test 

Solubility test was carried out in 20 mL of simulated gastric fluid or simulated intestinal fluid at 37 °C and 100 strokes per minute using cool bath shaker (ML-10F, Taitec Corporation, Saitama, Japan). The simulated gastric fluid was a 0.1-M HCL solution while the simulated intestinal fluid was prepared as previously reported [24]. Briefly, a mixed solution was prepared by completely dissolving 6 mM sodium taurocholate and 1.5 mM lecithin (COATSOME^®^ NC-50). Total of 7.8 g of KH_2_PO_4_ and 15.4 g of KCl were added to the mixed solution and the total volume was adjusted to 1L with distilled water in a volumetric flask. The pH of the final solution was adjusted to 6.5 using 1 M NaOH. Freeze-dried powders prepared with mannitol were added to 10 mL of dissolution media as a 5 mg of CoQ10 content. One milliliter of samples was removed after 24 h and filtered through a 0.45 µm PTFE-filter. The concentration of CoQ10 was determined by the HPLC method described in Section 2.6. 

### 2.11. Animal Study

All animal experiments were approved by the Animal Research Committee of Osaka University of Pharmaceutical Sciences and were performed according to the Institutional Committees’ regulations on animal experimentation. Male Sprague Dawley rats (9 weeks; 250–300 g; Japan SLC Inc., Shizuoka, Japan) were used. Six rats were used in each sample. The rats had fasted for 12 h before the start of the experiments. Rats were anesthetized using isoflurane. Untreated CoQ10 powder was dispersed in 0.5% carboxymethyl cellulose sodium salt. Untreated CoQ10 powder and CoQ10-loaded emulsion (equivalent to 30 mg/kg CoQ10 content) were orally administered to rats using an oral dosing syringe. Blood samples (400 μL) were taken from the jugular vein at predetermined time intervals following administration. Plasma was obtained from blood samples by centrifugation for 10 min at 9730*g*. 2-propanol (400 μL) was added to plasma (100 μL) and the mixture was vortexed for 5 min prior to centrifugation at 9730× *g* for 10 min to separate the plasma proteins. The supernatant (400 μL) was dried in a vacuum desiccator for one day and the residue was reconstituted with 100 μL of methanol. The CoQ10 concentration in the supernatant was measured by an HPLC system (e2695 and 2489; Waters, Milford, USA) with a COSMOSIL 5C18-MS-II column (5 μm, 150 mm × 4.6 mm, Nacalai). The mobile phase was ethanol and methanol (70:30, v/v), the flow rate was 0.7 mL/min, and the injection volume was 100 μL. CoQ10 was eluted at 40 °C and quantified at 275 nm. The area under the plasma concentration-time curve (AUC) was determined by the trapezoidal method. 

### 2.12. Statistical Analysis

The results of the animal study are expressed as mean ± S.E. of six experiments. Statistical significances between groups were analyzed using the Student t-test. Statistical significance was indicated as ** *p* < 0.01.

## 3. Results and Discussion

### 3.1. Evaluation of the Emulsion Without CoQ10

#### 3.1.1. Evaluation of RGSLs, Polysorbate 80, and RGSLs/Polysorbate 80 Particles

Particles of RGSLs (0.01, 0.1, or 1 mg), Polysorbate 80 (210 mg), or Polysorbate 80 (210 mg) combined with RGSLs (50 mg) were prepared by adding 8 mL of distilled water after evaporation. Figure 2 shows the particle size distribution of RGSLs, Polysorbate 80, and Polysorbate 80/RGSLs particles.

As shown in Figure 2a, when 0.01 mg of RGSLs was dispersed in 8 mL of distilled water after evaporation, the mean particle size of RGSLs was ca. 110.0 nm. These particles were identified as liposomal particles as sphingolipids are components of the cell membrane in the lipid bilayer and can be used to form the liposomal structure with phospholipids [25]. Aggregated particles and their precipitation were observed in the aqueous solution with more than 0.1 mg RGSLs. The particle prepared with polysorbate 80 and TG showed polydispersity as shown in Figure 2b. The peak of around 10 nm and the other peak were identified as polysorbate 80 micelles and RGSLs particles, respectively. The particles prepared with other ratios of polysorbate 80 and RGSLs also displayed polydispersity (data not shown). These results indicate that the bi-component of polysorbate 80 and RGSLs could not form mono-dispersed particles.

#### 3.1.2. Evaluation of the TG/RGSLs/Polysorbate 80 Emulsion

It was difficult to disperse the high RGSLs concentration in water because of its poor solubility as shown in Figure 2a. RGSLs are amphiphilic molecules with their hydrophobic component consisting of a lipid bilayer in biological membranes. Therefore, RGSLs could form an emulsion because of their amphiphilicity. Polysorbate 80 and TG were used as emulsifier and oil, respectively. The mean particle size of the emulsion as a function of RGSL concentration is shown in Figure 3. 

The emulsions were prepared with polysorbate 80 (210 mg), TG (70, 100, or 140 mg), and RGSLs by adding 8 mL of distilled water after evaporation. When the emulsions were prepared with the bi-components of polysorbate 80 and TG, the increase in TG amount in the emulsion enlarged the particle size. Mean particle sizes of the TG/polysorbate 80 (70/210, w/w), TG/polysorbate 80 (100/210, w/w), and TG/polysorbate 80 (140/210, w/w) emulsions were 13.7 nm, 42.7 nm, and 228.9 nm, respectively. The addition of RGSLs to the bi-component of TG/polysorbate 80 (100/210, w/w) and TG/polysorbate 80 (140/210, w/w) allowed preparation via self-emulsifying the microemulsion with a mean particle size of ca. 15 nm (Figure 4).

Some reports have indicated that lecithin can form nano-sized lipid emulsion by acting as the emulsifier [26,27]. Hence, a lipid emulsion formed by lecithin has been applied to treat various diseases [27,28]. RGSLs can assist in the formation of a microemulsion by self-emulsification. This indicates that RGSLs can act as an auxiliary emulsifying agent in TG/polysorbate 80 formulation. Increasing the RGSLs content provided a new interfacial area associated with a decrease in interfacial tension between the water and oil phase, thereby leading to particle size reductions. In contrast, an excessive addition of RGSLs enlarged the particle size. The emulsion prepared with 70 mg of TG increased the particle size because of the addition of fewer RGSLs compared to that with 100 and 140 mg of TG. The increase in particle size was considered as a formation of the RGSLs liposomal particle alone after RGSL’s orientation to the oil phase was saturated.

#### 3.1.3. Effect of Surfactant Content and Water Volume on Emulsion Formation

A microemulsion can be prepared by adding RGSLs to a mixture of TG and polysorbate, and then, dispersing them in water. Oil volume was an important factor for directing RGSLs to the interface between water and oil. The optimized formulation was further investigated based on surfactant content and water volume. Figure 5a shows the effect of polysorbate 80 content on the mean particle size of the emulsion.

The emulsions were prepared with polysorbate 80, TG (100 mg), and RGSLs (50 mg) with 8 mL of distilled water after evaporation because this formulation ratio showed the lowest particle size and the effect of addition of polysorbate 80 and water on particle size can be observed. The mean particle size of the emulsion decreased when polysorbate 80 content increased. The addition of more than 210 mg of polysorbate 80 resulted in the formation of a microemulsion with a mean particle size of ca. 15 nm. Figure 5b shows the effect of water volume on the particle size of the emulsion. The emulsions were prepared with polysorbate 80 (210 mg), TG (100 mg), and RGSLs (50 mg) and distilled water after evaporation. Increasing the water volume resulted in a decrease in the particle size of the emulsion. Microemulsion with a mean particle size of ca. 15 nm was prepared by adding more than 4 mL of water. Polysorbate 80 (210 mg), TG (100 mg), RGSLs (50 mg), and distilled water (8 mL) were selected as the formulation for the preparation of the CoQ10-loaded emulsion because emulsion prepared with this formulation showed the lowest particle size in the conditions investigated in this study.

### 3.2. Evaluation of CoQ10-Loaded Microemulsion

#### 3.2.1. Preparation of CoQ10-Loaded Emulsion

Emulsions containing CoQ10 (10–40 mg) were prepared with polysorbate 80 (210 mg), TG (100 mg), and RGSLs (50 mg) with 8 mL of distilled water after evaporation. The particle size of the CoQ10 loaded-emulsion was then determined (Figure 6).

When more than 20 mg of CoQ10 was added to the emulsion, an increase in particle size was observed. This increased particle size was considered to reflect the formation of only the CoQ10 nanoparticle after being saturated in the oil phase or swelling of the emulsion by the encapsulation of excess CoQ10. Therefore, the emulsion containing 20 mg of CoQ10 was prepared and evaluated as a change in mean particle size and encapsulation efficiency after storage for 4 weeks at 37 °C (Table 1).

The mean particle size of the CoQ10-loaded microemulsion was 16.83 ± 0.32 nm. The encapsulation efficiency of CoQ10 in the emulsion was almost 100%, indicating that CoQ10 was completely dissolved in the oil phase of the emulsion. When CoQ10-loaded microemulsion was stored for 4 weeks at 37 °C, the emulsion was stable without phase separation because a change in particle size and encapsulation efficiency did not occur compared to the product immediately after preparation. Normally, O/W emulsion is thermodynamically unstable because it is a colloidal dispersion of oil droplets in aqueous media. Long-term storage of O/W emulsion caused a phase separation of oil droplets from the aqueous media, resulting in an increase in particle size and compound leakage from the oil phase. A microemulsion is generally defined as a thermodynamically stable isotropic liquid consisting of oil, surfactant, and water [29]. The free energy of the oil droplets in the microemulsion system was lower than that of the separate phases (oil and water), indicating that the microemulsion was thermodynamically stable. The microemulsion prepared with RGSLs showed a moderate storage stability based on particle size and encapsulation efficiency.

#### 3.2.2. Preparation of Freeze-Dried Powder Containing CoQ10-Loaded Microemulsion

Because the freeze-dried powder could not be formed without a cryoprotectant, the CoQ10-loaded microemulsion was freeze-dried with a cryoprotectant of mannitol and sucrose. The morphology of freeze-dried particles was then evaluated. The morphology of the CoQ10 powder, RGSLs powder, and the freeze-dried particles was observed using SEM (Figure 7).

RGSLs and CoQ10 powders showed aggregates of particles that appeared formless and plate-like, respectively. The sucrose and mannitol powders displayed particles with a smooth surface and rough surface, respectively. Freeze-dried particles with sucrose showed an irregular particle with a rough surface. Aggregation of the fine particle was observed in the freeze-dried particles with mannitol. Particle size distribution after re-dispersal of the freeze-dried particles is shown in Figure 8.

The particle size distribution after re-dispersal of the freeze-dried particles with mannitol was almost the same as that before freeze-drying. However, when the freeze-dried particles were prepared with sucrose, an increase in particle size was observed after the re-dispersal of the particles. Sugar alcohols such as sucrose and mannitol are widely used as a cryoprotectant to improve the re-dispersibility of emulsions in the freeze-drying process [30]. The presence of sugar alcohol decreases the freezing temperature of water, and then increases the amount of non-freezing water for oil droplet dispersal. For the concentrated sugar alcohol solution in the freeze-drying process, a close interaction does not exist with the oil droplets in non-freezing water, thereby inhibiting the merging of the oil droplets. In this study, the re-dispersibility of the emulsion after freeze-drying could, therefore, be enhanced by the addition of mannitol. 

The solubility of CoQ10 from the freeze-dried particles of the CoQ10-loaded microemulsion with mannitol was compared to that of the CoQ10 powder in simulated gastrointestinal fluid (Table 2).

CoQ10-loaded microemulsion showed a dramatically higher solubility than the CoQ10 powder in simulated gastrointestinal fluids. CoQ10 is a highly lipophilic compound that exhibits an extremely low solubility in simulated gastrointestinal fluids as shown in Table 2. Nonetheless, the CoQ10-loaded microemulsion induced by RGSLs showed enhanced solubility in simulated gastrointestinal fluids, indicating that more than 500 µg/mL CoQ10 can be dissolved in emulsion with a formulation of RGSL (50 mg), polysorbate 80 (210 mg), and TG (100 mg). This result suggests that CoQ10 could be stably encapsulated in the oil phase even in the stomach and small intestine with a different pH value and salt concentration.

#### 3.2.3. Oral Absorption of CoQ10 from CoQ10-Loaded Microemulsion

Figure 9 shows the plasma concentration–time profiles of CoQ10 in rats after being orally administered CoQ10 powder and CoQ10-loaded microemulsion. 

The oral absorption of CoQ10 was dramatically enhanced by encapsulating CoQ10 into the emulsion. Maximum drug concentrations (Cmax) of the CoQ10 powder and CoQ10-loaded microemulsion were 84.23 ± 17.79 and 578.09 ± 47.47 ng/mL, respectively (Table 3). For the CoQ10-loaded microemulsion, the increase in the AUC was 7.1-fold higher than that of the CoQ10 powder. Some reports have discussed the enhancement of CoQ10 oral absorption [31,32]. Encapsulating CoQ10 into solanesyl poly(ethylene glycol) succinate micelle and decreasing the particle size of its crystal improved its oral absorption. Although CoQ10 displays an extremely low solubility, these improved the oral absorptions caused by enhanced dissolution and solubility in the processing technologies. CoQ10-loaded microemulsion induced by RGSLs showed higher CoQ10 solubility in simulated gastrointestinal fluids. Enhanced solubility because of emulsification could contribute to the increased oral absorption of CoQ10. Its release from the microemulsion in the small intestine is also very important for its oral absorption, indicating that CoQ10 molecules must be released on the surface of the epithelial membrane before membrane permeation. Digestion by enzymes such as lipase is one of the processes for a compound’s release from an emulsion [33]. An oral absorption study suggested that microemulsion formed using RGSLs could be digested in the gastrointestinal tract of rats and CoQ10 could be released, thereby contributing to its enhanced oral absorption. 

## 4. Conclusions

RGSLs can assist in emulsifying the formulation consisting of polysorbate 80 and TG. RGSLs can exist on the interface of the water and oil phase and provide a new interfacial area by decreasing the interfacial tension between the oil and water phase. As a result, adding RGSLs to the polysorbate 80/TG formulation induced a reduction in the particle size of the emulsion. Prepared CoQ10-loaded microemulsion displayed high storage stability based on the particle size and encapsulation efficiency of CoQ10. Based on the high re-dispersibility of the CoQ10-loaded microemulsion powder, it could be prepared by freeze-drying with a cryoprotectant. Its freeze-dried particles showed enhanced CoQ10 solubility in simulated gastrointestinal fluids compared to the CoQ10 powder. Furthermore, oral administration of CoQ10-loaded microemulsion to rats significantly increased Cmax and AUC of CoQ10 compared to the values achieved with the CoQ10 powder. Altogether, the results demonstrate that food grade microemulsion induced by RGSLs will be an effective processing technique to improve the solubility and absorption of CoQ10.

## Figures and Tables

**Figure 1 foods-08-00502-f001:**
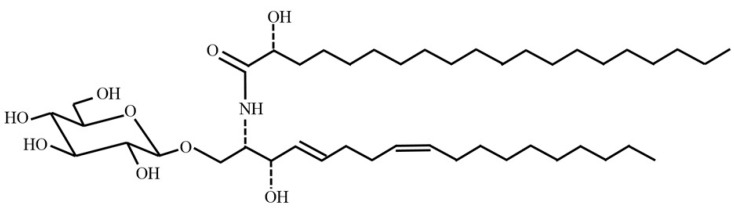
A chemical structure of rice glycosphingolipids. The main composition of the fatty acid and sphingoid base in glycosphingolipids (RGSLs) was C20:0 and d18:2, respectively.

**Figure 2 foods-08-00502-f002:**
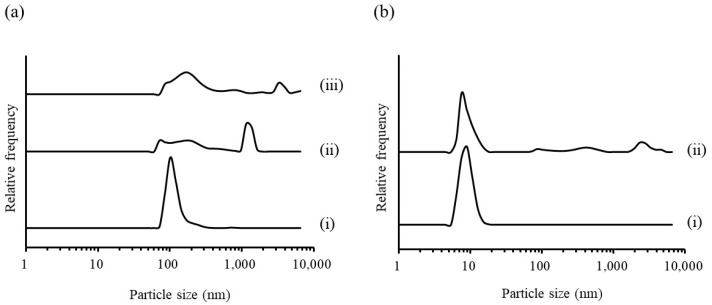
(**a**) Particle size distribution of rice glycosphingolipids (RGSLs) alone. ((i) 0.01 mg, (ii) 0.1 mg, and (iii) 1 mg in 8 mL of distilled water). (**b**) Particle size distribution of (i) Polysorbate 80 alone or (ii) mixture of Polysorbate 80/RGSLs in 8 mL of distilled water.

**Figure 3 foods-08-00502-f003:**
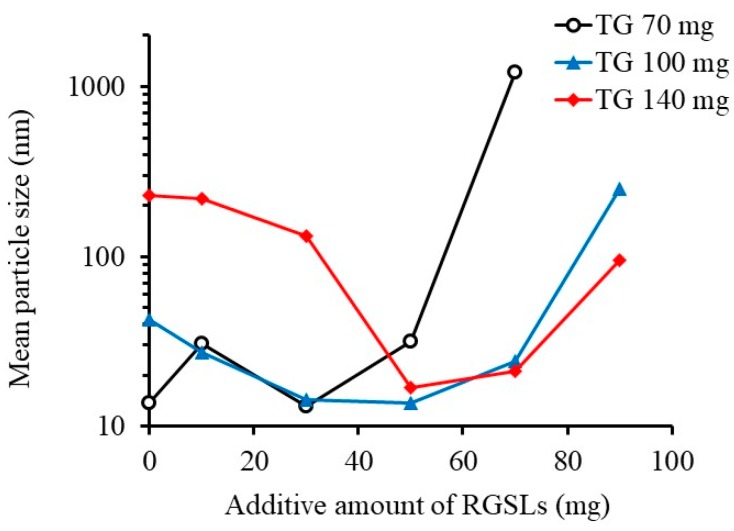
Effect of the additive amount of rice glycosphingolipids (RGSLs) in the formulation of polysorbate 80/caprylic/capric triglyceride (TG) on the particle size of the emulsion. The results are expressed as averages of three experiments. The emulsions were prepared with polysorbate 80 (210 mg), TG (70, 100 or 140 mg) and RGSLs by adding 8 mL of distilled water after evaporation.

**Figure 4 foods-08-00502-f004:**
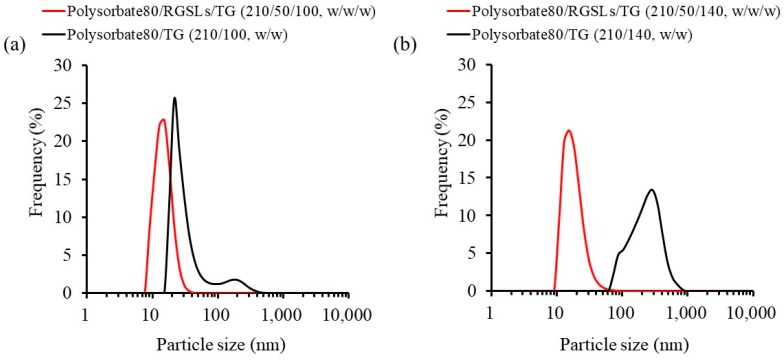
Particle size distribution of the emulsion with or without rice glycosphingolipids (RGSLs). (**a**) The emulsions were prepared with polysorbate 80 (210 mg), TG (100 mg), and RGSLs (0 or 50 mg). (**b**) The emulsions were prepared with Polysorbate 80 (210 mg), TG (140 mg) and RGSLs (0 or 50 mg).

**Figure 5 foods-08-00502-f005:**
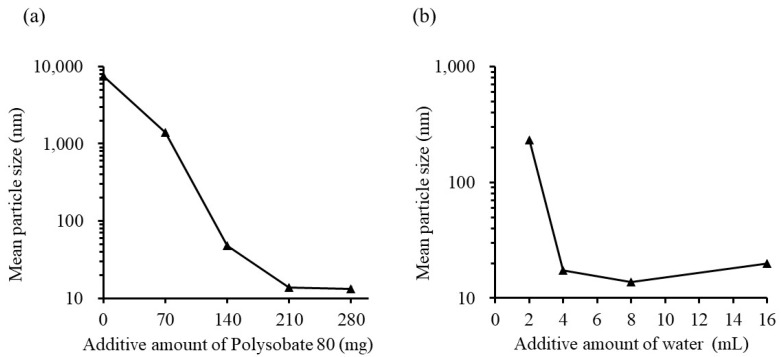
(**a**) Effect of the additive amount of polysorbate 80 on the particle size of the emulsion. The emulsions were prepared with polysorbate 80, TG (100 mg) and RGSLs (50 mg) by adding 8 mL of distilled water after evaporation. (**b**) Effect of the additive amount of water on the particle size of the emulsion. The results are expressed as an average of three experiments. The emulsions were prepared with polysorbate 80 (210 mg), TG (100 mg), and RGSLs (50 mg) by adding distilled water after evaporation.

**Figure 6 foods-08-00502-f006:**
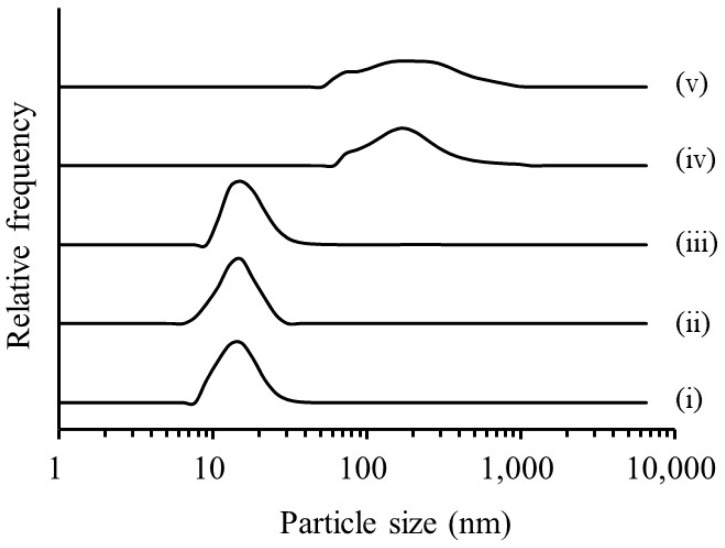
Effect of the additive amount of coenzyme Q10 (CoQ10) on particle size of the emulsion. (Additional CoQ10 content; (i) 0 mg, (ii) 10 mg, (iii) 20 mg, (iv) 30 mg and (v) 40 mg).

**Figure 7 foods-08-00502-f007:**
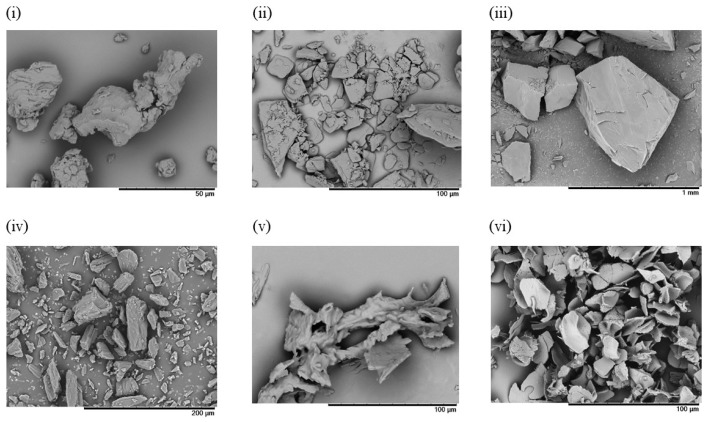
Scanning electron microscope (SEM) images of (**i**) coenzyme Q10 (CoQ10) powder, (**ii**) rice glycosphingolipid (RGSLs) powder, (**iii**) sucrose powder, (**iv**) mannitol powder, (**v**) freeze-dried particles of CoQ10-loaded microemulsion with sucrose, (**vi**) freeze-dried particles of CoQ10-loaded microemulsion with mannitol.

**Figure 8 foods-08-00502-f008:**
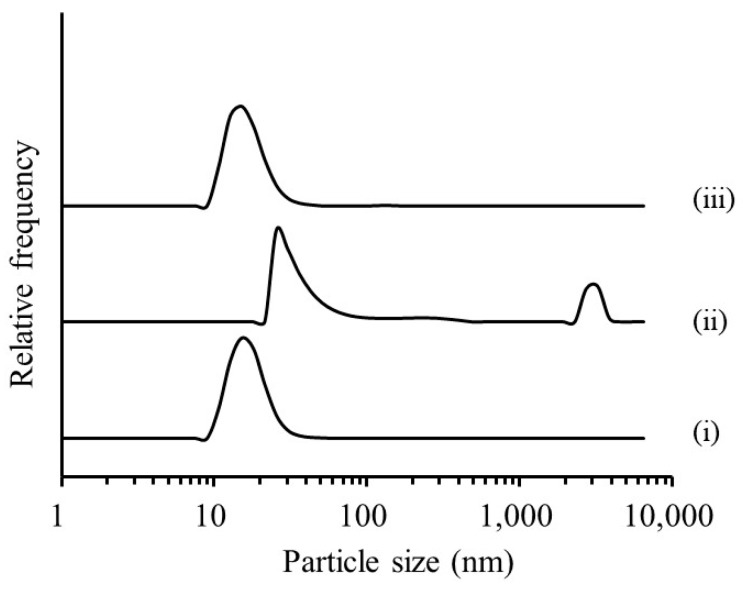
Particle size distribution of (i) emulsion before freeze drying, (ii) re-dispersed emulsion after freeze drying with sucrose, and (iii) re-dispersed emulsion after freeze drying with mannitol.

**Figure 9 foods-08-00502-f009:**
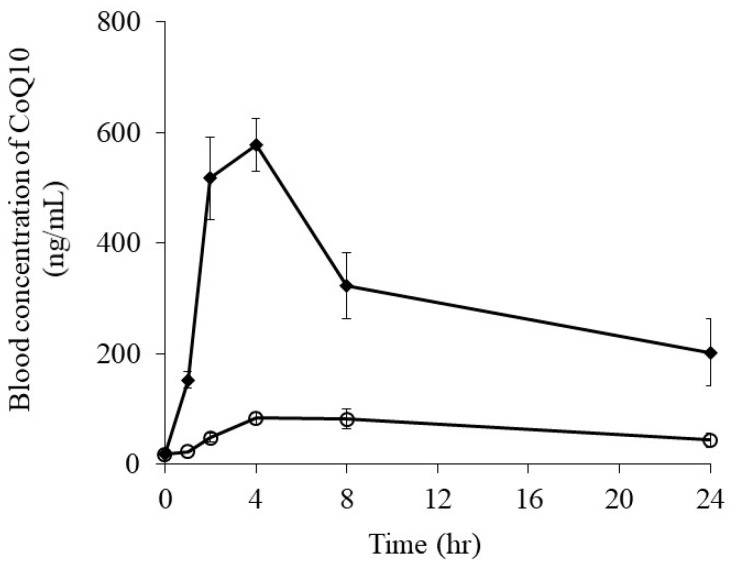
Plasma concentration–time profiles of coenzyme Q10 (CoQ10) in rats after oral administration of CoQ10 powder and CoQ10-loaded microemulsion (〇; CoQ10 powder, ◆; CoQ10-loaded microemulsion). Each point represents mean ± S.E. (*n* = 6).

**Table 1 foods-08-00502-t001:** Changes in particle size and encapsulation efficiency of coenzyme Q10 (CoQ10) loaded-microemulsion after storage for 4 weeks at 37 °C. Each point represents the mean ± SD (*n* =3).

Storage Time (week)	Mean Particle Size (nm)	Encapsulation Efficiency (%)
0	16.83 ± 0.32	100.34 ± 0.92
1	17.37 ± 0.90	98.71 ± 6.66
2	16.97 ± 1.40	102.3 ± 4.86
4	17.17 ± 1.81	105.80 ± 4.42

**Table 2 foods-08-00502-t002:** Solubility of coenzyme Q10 (CoQ10) from the CoQ10 powder and freeze-dried particles of CoQ10-loaded microemulsion with mannitol in simulated gastrointestinal fluids.

Test Solution	Solubility of CoQ10 (μg/mL)	
	CoQ10 Powder	CoQ10-Loaded Microemulsion
Simulated gastric fluid	N.D.	507.31 ± 54.00
Simulated intestinal fluid	0.44 ± 0.03	455.57 ± 51.91

**Table 3 foods-08-00502-t003:** Pharmacokinetic parameters of coenzyme Q10 (CoQ10) after oral administration of CoQ10 powder and CoQ10-loaded microemulsion to rats (** *p* < 0.01 compared to CoQ10 powder). Each point represents mean ± S.E. (*n* = 6).

	CoQ10 Powder	CoQ10 Loaded-Microemulsion
Cmax (ng/mL)	84.2 ± 17.8	578.1 ± 47.5
Tmax (h)	4.0	4.0
AUC0-24 h (ng·h/mL)	1050.9 ± 202.6	7503.6 ± 1079.3 **

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
