# Peer review of "Formation of Food Grade Microemulsion with Rice Glycosphingolipids to Enhance the Oral Absorption of Coenzyme Q10"

_foods, 2019, doi:10.3390/foods8100502_

Round 1

Reviewer 1 Report

The authors have addressed all comments from the reviewers, and corrected the manuscript accordingly.

The manuscript would however, still benefit from another round of minor corrections, and also, I still do not think the authors have determined the saturation solubility of CoQ10 in HCl and FaSSIF when administrated as freeze-dried powder.   

As commented last time, in the description of the solubility study is stated that “Freeze-dried powders prepared with mannitol were added to 10 mL of dissolution media as a 5 mg of CoQ10 content” (l.149-151). This corresponds to 500µg/mL which is identical to the claimed solubility. From these results, I do not believe that the saturation solubility has been reached, and the data should NOT be presented as such. As the solubility of CoQ10 from the freeze-dried particles of the Co-Q10-loaded microemulsion, will depend on the amount of emulsion added to the aqueous media, the study is nonsense. The solubility of CoQ10 in a fix emulsion composition, i.e with the empty microparticles (no CoQ10) in a fix amount of media, can be measured med adding a surplus of CoQ10 and stir the mixture till it reaches the saturation solubility (equilibrium between solid and solubilized CoQ10). However, as stated previously, the solubility will depend on the amount of water used to prepare the emulsion, and so this ratio needs to be physiologically relevant before the data matters.

-          With the present results, it can be stated that more than 500 µg/mL CoQ10 can be dissolved in simulated gastrointestinal media when administrated with xx amounts of RGSL, Polysorbate 80, and TC.

L.15, please edit the sentence. CoQ10 is not dissolved in the freeze-dried solid particle, and you have not measured the solubility in a mixed emulsion mix, or in the lipid-preconcentrate.

L.31 Change “inhibited” with “limited”

L.33-36, after editing, the sentence does not make sense. Split in two.

L.42, The abbreviate HLB needs to the explained.

L.49-50, Toxicity is still NOT a defining characteristic of an emulsifier. Not all low toxicity compounds are emulsifiers. Please edit.

L.63. Please add a reference to support the statement

L.284. Please change the word “high”, as four weeks cannot be considered high storage stability

Author Response

Dear Editor,

Thank you for your letter and the reviewers’ comments with respect to our manuscript entitled “Formation of food grade microemulsion with rice glycosphingolipids to enhance the oral absorption of Coenzyme Q10” (Foods-547500). We have considered their comments carefully and have made corrections which we hope to meet with their approval. The final check of English language and style were performed by native English speaker. The revised parts are highlighted in yellow color in the manuscript. 

Response to Reviewer 1 Comments

Point 1: As commented last time, in the description of the solubility study is stated that “Freeze-dried powders prepared with mannitol were added to 10 mL of dissolution media as a 5 mg of CoQ10 content” (l.149-151). This corresponds to 500µg/mL which is identical to the claimed solubility. From these results, I do not believe that the saturation solubility has been reached, and the data should NOT be presented as such. As the solubility of CoQ10 from the freeze-dried particles of the Co-Q10-loaded microemulsion, will depend on the amount of emulsion added to the aqueous media, the study is nonsense. The solubility of CoQ10 in a fix emulsion composition, i.e with the empty microparticles (no CoQ10) in a fix amount of media, can be measured med adding a surplus of CoQ10 and stir the mixture till it reaches the saturation solubility (equilibrium between solid and solubilized CoQ10). However, as stated previously, the solubility will depend on the amount of water used to prepare the emulsion, and so this ratio needs to be physiologically relevant before the data matters.

-          With the present results, it can be stated that more than 500 µg/mL CoQ10 can be dissolved in simulated gastrointestinal media when administrated with xx amounts of RGSL, Polysorbate 80, and TC.

Response 1:  Thank you for your kind advice. According to the reviewer’s suggestion, we changed the sentence (page10, line 324-325)

--- , indicating that more than 500 µg/mL CoQ10 can be dissolved in emulsion with a formulation of RGSL (50 mg), Polysorbate 80 (210 mg), and TG (100 mg).

Point 2:   L.15, please edit the sentence. CoQ10 is not dissolved in the freeze-dried solid particle, and you have not measured the solubility in a mixed emulsion mix, or in the lipid-preconcentrate.

Response 2:  As you know, if CoQ10 are released from oil phase and precipitated in freeze-dried powder during freeze drying process, re-dispersed emulsion shows a low encapsulation efficiency of CoQ10 and an increasement of particle size. On the other hand, the encapsulation efficiency of CoQ10 and particle size distribution after re-dispersal of the freeze-dried particles was almost the same as that before freeze drying, indicating that “CoQ10 loaded-microemulsion was also successfully powdered after freeze-drying with a cryoprotectant.”

Therefore, this sentence is correct.

Point 3:   L.31 Change “inhibited” with “limited”

Response 3:  Thank you for your comments. According to the reviewer’s suggestion, we changed the sentence form “inhibited” to “limited” (page1, line 31)

Point 4:  L.33-36, after editing, the sentence does not make sense. Split in two.

Response 4: According to the reviewer’s suggestion, we changed the sentence. (page1, line 33-35)

Point 5:   L.42, The abbreviate HLB needs to the explained.

Response 5: Thank you for your comments. According to the reviewer’s suggestion, we have added the abbreviate of HLB (page1, line 41-42)

Point 6:   L.49-50, Toxicity is still NOT a defining characteristic of an emulsifier. Not all low toxicity compounds are emulsifiers. Please edit.

Response 6: Thank you for your comments. According to the reviewer’s suggestion, we have changed the sentence. (page1, line 48-49)

Point 7:   L.63. Please add a reference to support the statement

Response 7: We have described in L.63 as “Only a few studies have reported the application of RGSLs as an emulsifier.” Reported study is patent. Therefore, we did not add in references.

Point 8:   L.284. Please change the word “high”, as four weeks cannot be considered high storage stability

Response 8: Thank you for your comments. According to the reviewer’s suggestion, we have changed the word (page8, line 287)

Reviewer 2 Report

Thanks to the Authors for the given explanations and introduced corrections.

Author Response

The final check of English language and style were performed by native English speaker. 

This manuscript is a resubmission of an earlier submission. The following is a list of the peer review reports and author responses from that submission.

Round 1

Reviewer 1 Report

The authors presented another problem regarding delivery system to encapsulate lipophilic substrate  into O/W emulsion. They describe application of RGSL as emulsifier to improve oral absorption of coenzyme Q10 using Polysorbate 80/TG microemulsion. The Authors presented nicely done chemical work with description of each step of preparations. However, the presentations of results is very scarce, the descriptions of used methods also need improvements.

The Authors based their assumptions mostly on DLS method, however, they did not define what type of average weighted radius are presented in figures. There are three average weighted presentations of results and their interpretations are different.  This should be added in Methods section. I think that polidyspersity indexes for the measured samples should also be collected in Table. To get more information from DLS I would suggest to measure zeta potentials.  

I have also some comments questions regarding presented results.

Figure 1a shows DLS distribution, I wonder at 1 mg was the sample diluted for measurements or not. Was this sample opaque or transparent?  This question also relates to all presented DLS measurements. Establishing this knowledge we may discuss presented results.

The applied methods and presented results, sems to me, indicate that depending on the conditions the Authors obtained mixed micro- nano- emulsions or presence of other heterogeneous phases.

Presented stability does not indicate unambiguously that microemulsion was formed. Especially when conditions were constant. There is also kinetic stability which depends on some physic-chemical factors.

Figure with Legend should be self explaining part of manuscript, thus, please, give more information in the Legends to presented figures. In legend to figure 3 the concentrations of RGSL should be added. This same in figure 4.

100% efficiency of loading. Impressive result. I can not find what was the loading efficency without RGSL.Would you indicte this value.  Did you try evaluate the number of CoQ10 per micelle?

In my opinion Supplementary Figure 1 does not introduce to much knowledge to presented problem. The plot is crowded and is dominated by crystalline forms of added sugars thus it may be omitted.

Reviewer 2 Report

Review: Uchiyama H et al. Formation of food grade microemulsion with rice glycosphingolipids to enhance the oral absorption of Coenzyme Q10

The manuscript by Cao et al. describes the design of a microemulsion containing rice glycosphingolipids for increased absorption of Coenzyme Q10. The article reads well and is logically structured, however, the main purpose of the study is somewhat unclear. The following points should be addressed:

Major corrections:

L.62-63. Please consider the purpose of the study and rewrite the abstract and introduction accordingly. Here you state the purpose is the purpose of the study was to clarify the possible use of RGSLs as an emulsifier and the effect of RGSLs addition to emulsion formation. I.e. the focus is on RGSL, and CoQ10 is only mentioned as a model compound. However, in the abstract and the first part of the introduction, the focus is in CoQ10, and the purpose appear to be, to design a formulation to ensure high oral bioavailability..

Depending on, what is presented as the main purpose, you could consider showing a structure of RGSL

For the optimization of the emulsion formation a design of experiment (DoE) approach would have been appropriate. And ternary diagrams showing the effect of multiple factors at the same time, a better way of showing the results.

Have you considered the effect of GI-fluids or co-administrated water on the dispersion of the lipid concentrate, or prepared microemulsion? In section 3.1.3. you evaluate the necessary amount of water to form an emulsion with a small particle size. However, following oral administration liquid will be available in the stomach due to co-administration of water and gastric secretions. It would be an interesting study to evaluate the impact of in vivo relevant amount of aqueous media on the dispersion of the TC/RSGL/Polysorbate80 mixture.

In section 3.2.2. you discuss the solubility of the CoQ10 in simulated gastric and intestinal media,  stating that the result suggests that CoQ10 could be stably encapsulated in the 304 oil phase even in the stomach and small intestine with a different pH value and salt concentration. However, it is not clear if the tested dose/volume ratio is representative of the in vivo situation. Furthermore, in the description of the solubility study is stated that “Freeze-dried powders prepared with mannitol were added to 10 mL of dissolution media as a 5 mg of CoQ10 content” (l.136-137). This corresponds to 500µg/mL which is identical to the claimed solubility. This makes me think, that the saturation solubility has not been reached, and the data cannot be trusted as a solubility value.

What is the point of XRPD measurements conducted on the freeze-dried powder? Also, the description of the XRPD measurement method is missing. Delete in not relevant.

Minor corrections:

L.30. Delete “methanol” as it is not a relevant solvent for oral intake of CoQ10

L.32. Please state the molecular weight of CoQ10

L.34-50: Have all of these formulations been prepared with CoQ10. If not, please clearly specify, which formulations were made for CoQ10, and which for other compounds

L.49-50: I do not understand the sentence. What is the differences between an emulsifier and a surfactant? Also it is not, the low toxicity which characterizes it as a surfactant, it is the structure

L.51-61 Introduce the abbreviation GSL at its first mentioning, and be consistent with its use.

L.60-61 please be more precise in the description of the emulsion, e.g. o/w emulsion containing CoQ10.

L.80-83.In how many replicates was the particle size measurements performed?

L.85 Also please state the reason for testing different amount of RGSL.

L.103 What is in the supernatant? The lipid phase? I assume that centrifugation led to a phase separation of an aqueous phase and a lipid phase..? However, please state this clearly.

L.136, please state the content of the simulated intestinal fluid to ease the reading

L.140 Please state how many rats (in each group) were used for the study.

L.85 and l.165, and l.170 does not match. In the method section it is stated that particles are mad from RGSLs (0.01, 0.1, and 50 mg), however in the results section it says RGSLs (0.01, 0.1, or 1 mg), which is correct?!

L.187-188 please re-write to form one sentence.

L.189: please change to: Figure 2 show the mean particle size of the emulsions as a function of RGSL concentration. Please also state that the amount of TG was varied.

Figure 2: Figure (consider to) use a ternary phase diagram instead to get a better overview of the effects of TG and RGSL. ALSO please correct the title below the x-axis to RGSL

Figure 3. Legend. Please specify what is depicted in a) and b).

L.215: Please write clear and complete sentences:  A microemulsion could be prepared by adding RGSLs to a mixture of TG and polysorbate, and dispersing it in water.

Figure 4. Please state the composition of the emulsion to which Polysorbate 80 and/or water was added. And again if you had varied

L.224: Please state clearly the reason for chosen this formulation: Polysorbate 80, TG (100 mg), and RGSLs (50 mg)

L.232: Again, please clearly state the reason for your formulation selection.

L.242: When more than 20 mg of CoQ10 was added..

Table 1. How many replicates is the data representing? And is is mean±SD?

L.253: Please state how you evaluated the possibility of phase separation. I assume it is based on the particle size evaluation, but I should not be in doubt when I am reading the paper.

L-264: What powder? Please be specific

L.265: .. such as mannitol and sucrose. Please edit to: .. in this case, mannitol and sucrose.

L.269: Hmmm, I do not I agree with this statement, when freeze-dried with Sucrose, peaks are visible at 19 degrees, which does not appear to come from sucrose but rather from CoQ10. And what is the peak at 10 degrees following freeze-drying with mannitol? It does not appear to correlate with either the crystal peaks of mannitol, RGSL or CoQ10..?! (maybe this is what you state in L.269-270, however, not very clearly!)

Figure 7: Please be consistent in the order in which you describe the different formulations. Succrose has been mentioned before mannitol up till here, where you changed the order.

L.300-301: Please complete the sentence; “…showed a dramatically higher solubility..” by specifying the media.

Table 3: Please correct the typo in the last column of the table: CoQ10 loaed should be CoQ10 loaded

L319-320. Please add a reference to the statement.